# DEFactor: Differentiable Edge Factorization-based Probabilistic Graph Generation

## Abstract

Generating novel molecules with optimal properties is a crucial step in many industries such as drug discovery. Recently, deep generative models have shown a promising way of performing de-novo molecular design. Although graph generative models are currently available they either have a graph size dependency in their number of parameters, limiting their use to only small graphs or are formulated as a sequence of discrete actions needed to construct a graph, making the output graph non-differentiable w.r.t the model parameters, therefore preventing them to be used in scenarios such as conditional graph generation. In this work we propose a model for conditional graph generation that directly optimises properties of the graph, and generates a probabilistic graph, making the decoding process differentiable. We demonstrate favourable performance of our model on prototype-based molecular graph conditional generation task.

## 1 Introduction

In this paper we address the problem of learning probabilistic generative graph models for tasks such as the conditional generation of molecules with optimal properties. More precisely we focus on generating realistic molecular graphs, similar to a target molecule (the prototype).

The main challenge stems from the discrete nature of molecules. This particularly prevents us from using global discriminators that assess generated samples and back-propagate their gradients to guide the optimisation of the generator. This becomes very important in cases where we want to either optimise the property of a graph or explore the vicinity of an input graph (prototype) for conditional optimal generations, an approach that has proven successful in controlled image generation [Chen et al. (2016); Odena et al. (2016)].

A number of recent approaches aim to address this limitation by performing indirect optimisation (You et al., 2018a; Li et al., 2018a). You et al. You et al. (2018a) formulate the molecular graph optimisation task in a reinforcement learning setting, and optimise the loss with policy gradient Yu et al. (2016). However policy gradient tends to suffer from high variance during training. Kang and Cho Kang & Cho (2018) suggest a reconstruction-based formulation which is directly applicable to discrete structures and does not require gradient estimation. However, it is limited by the number of samples available. Moreover, there is always a risk that the generator simply ignores the part of the latent code containing the property that we want to optimise. Finally, Jin et al, Jin et al. (2018) utilise Bayesian optimisation to optimise a proxy (the latent code) of the molecular graph, rather than the graph itself.

In contrast, Simonovsky and Komodakis Simonovsky & Komodakis (2018) and De Cao and Kipf Cao & Kipf (2018) have proposed decoding schemes that output graphs (adjacencies and node/edge features tensors) in a single step, and so are able to perform direct optimisation on the probabilistic continuous approximation of a graph. However, both decoding schemes make use of fixed size MLP layers which restricts their use to very small graphs of a predefined maximum size.

In this work, we address these issues directly by:

- Proposing a novel probabilistic graph decoding scheme that is computationally efficient and capable of generating arbitrary sized graphs (Section 3).

- Evaluating the generator's capacity to modify molecular graph attributes in the context of prototype-based molecular graph generation (Section 4).

Our approach (DEFactor) depicted in Figure 2 aims to directly address these issues with a probabilistic graph decoding scheme that is end-to-end differentiable, computationally efficient w.r.t the number of parameters in the model and capable of generating arbitrary sized graphs. We evaluate DEFactor on the task of constrained molecule property optimisation Jin et al. (2018); You et al. (2018a) and demonstrate that our results are competitive with recent results.

## 2 Related work

**Lead-based Molecule Optimisation.** The aim here is to obtain molecules that satisfy a target set of objectives, for example activity against a biological target while not being toxic *or* maintaining certain properties, such as solubility. Currently a popular strategy is to fine-tune an pretrained generative model to produce/select molecules that satisfy a desired set of properties Segler et al. (2017).

Bayesian optimisation is proposed to explore the learnt latent spaces for molecules in GÃşmez-Bombarelli et al. (2016), and is shown to be effective at exploiting feature rich latent representations Kusner et al. (2017); Dai et al. (2018); Jin et al. (2018). In Li et al. (2018b;a) sequential graph decoding schemes whereby conditioning properties can be added to the input are proposed. However these approaches are unable to perform direct optimisation for objectives. Finally You et al. (2018a) reformulates the problem in a reinforcement learning setting, and objective optimisation is performed while keeping an efficient sequential-like generative scheme You et al. (2018b).

**Graph Generation Models.** Current work on graph generation can be divided into two type of graph decoding schemes. Sequential methods to graph generation [ You et al. (2018b); Li et al. (2018a); You et al. (2018a); Li et al. (2018b)] aim to construct a graph by predicting a sequence of addition/edition actions of nodes/edges. Starting from a sub-graph (normally empty), at each time step a discrete transition is predicted and the sub-graph is updated. Although sequential approaches enable us to decouple the number of parameters in models from the the maximum size of the graph processed, due to the discretisation of the final outputs, the graph is still non-differentiable w.r.t. to the decoder's parameters. This again prevents us from directly optimising for the objectives we are interested in.

In contrast to the sequential process Cao & Kipf (2018); Simonovsky & Komodakis (2018) reconstruct probabilistic graphs. These methods however make use of fixed size MLP layers when decoding to predict the graph adjacency and node tensors. This however limits their use to very small graphs of a pre-chosen maximum size. They therefore restrict study and application to small molecular graphs; a maximum number of 9 heavy atoms, compared to approximately 40 in sequential models.

We propose to tackle these drawbacks by designing a graph decoding scheme that is:

- **Efficient**: so that the number of parameters of the decoder does not depend on a fixed maximum graph size.

- **Differentiable**: in particular we would like the final graph to be differentiable w.r.t the decoder's parameters, so that we are able to directly optimise the graph for target objectives.

**Edge-factorization.** The idea of using tensor factorization methods to decode edges is not new and has been extensively used in relational inference tasks (Nickel & Tresp (2013)). Kipf & Welling (2016a) recently proposed an graph autoencoder for rich node embeddings

learning and link prediction. However their formulation suppose a fixed size graph in which we want to predict existing and potential novel links between its nodes. Such assumption is not applicable to the case of molecular graph optimization where given a property we want to be able to generate a graph of *a priori* unknown size. To that extent we suggest a novel autoregressive model for nodes embeddings generation which, when combined with edge-factorization graph decoding constitute our full graph decoder. We describe the model more extensively in the next section.

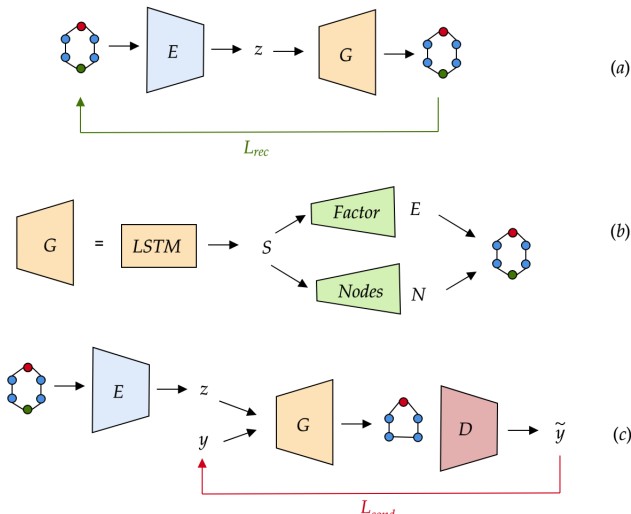

Figure 1: (a) is the full autoencoder (step **1** to **4**) , (b) is expanding the steps (**3** for the LSTM and **4** for the factorization and node decoding) of the generator $G$ of the autoencoder and (c) is the conditional setting a discriminator $D$ that assesses the outputs and gives its feedback to the generator $G$. $L_{rec}$ (resp. $L_{cond}$) refers to the reconstruction (resp. condtional) loss described in section 3.3.

## 3 DEFACTOR

Molecules can be represented as graphs $G = (V, E)$ where atoms and bonds correspond to the nodes and edges respectively. Each node in V is labeled with its atom type which can be considered as part of its features. Equally, a molecular graph can be defined by its adjacency tensor $E \in \{0, 1\}^{n \times n \times e}$ where $n$ is the number of nodes (atoms) in the graph and $e$ is the number of edge (bond) types that we can have between two atoms and the node types are represented by a node feature tensor $N \in \{0, 1\}^{n \times d}$ which is composed of several one-hot-encoded properties. In the following sections we will describe our proposed DEFactor model applied to molecular graphs.

### 3.1 GRAPH CONSTRUCTION PROCESS

Given a molecular graph defined as $G = (N, E)$ we propose to leverage the edge-specific information propagation framework described in Simonovsky & Komodakis (2017) to learn a set of informative embeddings from which we can directly infer a graph. More specifically, our graph construction process is composed of two parts:

- An **Encoder** that in
  - **step 1** performs several spatial graph convolutions on the input graph, and in
  - **step 2** aggregates those embeddings into a single graph latent representation.

- A **Decoder** that in

    - **step 3** autoregressively generates a set of continuous node embeddings conditioned on the learnt latent representation, and in

    - **step 4** reconstruct the whole graph in an edge-factorization fashion.

Figure 2 (a) and (b) provides a summary of those 4 steps.

**Steps 1 and 2: Graph Representation Learning.**   We propose an encoder that makes an efficient use of the information contained in the bonds by having a separate information propagation channel for each bond type. The information is propagated in the graph using a Graph Convolutional Network (GCN) update rule (Kipf & Welling, 2016b) so that each node embedding can be written as a weighted sum of the edge-conditioned information of its neighbors in the graph. Namely for each $l$-th layer of the encoder, the representation is given by:

$$H^l = \sigma(\sum_e [D_e^{-\frac{1}{2}} E_e D_e^{-\frac{1}{2}} H^{l-1} W_e^l] + D^{-1} H^{l-1} W_s^l), \tag{1}$$

where $E_e$ is the $e$-th frontal slice of the adjacency tensor, $D_e$ the corresponding degree tensor and $W_e^l$ and $W_s^l$ are learned parameters of the layer.

Once we have those node embeddings we further aggregate them to obtain a fixed-length latent representation of the graph. We propose to parametrize this aggregation step by an *LSTM* and we compute the graph latent representation by a simple linear transformation of the last hidden state of this **Aggregator**:

$$z = g_{agg}(f_{LSTM}^e(\{H^K\})). \tag{2}$$

Even though the use of an LSTM makes the aggregation non permutation-invariant, similar to GraphSAGE Hamilton et al. (2017), we adapt the LSTM aggregator to work on a randomly permuted set of embeddings and noticed that it did not affect the performance of the model.

In the subsequent steps we are interested in designing a graph decoding scheme from the latent code that is both scalable and powerful enough to model the interdependencies between the nodes and edges in the graph.

**Step 3: Autoregressive Embeddings Generation.**   As specified above, we are interested in building a graph decoding scheme that models the nodes and their connectivity (represented by continuous embeddings $S$) in an autoregressive fashion so that the latent code on which is conditioned the decoding has enough *dimensions* to encode more high-level features. Like stated previously, methods that suggest to compute the graph all at once (Simonovsky & Komodakis, 2018; Cao & Kipf, 2018) model each node and edge as conditionally independent given the latent code $z$: this means that every detail of their dependencies within the graph has to be encoded in this latent variable: that does not leave much room for high-level feature learning if the size of the latent code is not chosen carefully. We propose to tackle this drawback by autoregressive generation of the continuous embeddings $s = [s_0, s_1, ..., s_n]$ for $n$ nodes, with the embedding containing enough information about the node itself and its neighbourhood. More precisely we model the generation of node embeddings such that:

$$p(s|z) = \prod_{i=1}^{n} p(s_i|s_{<i}, z). \tag{3}$$

In our model, the autoregressive generation of embeddings is parametrized by a simple Long Short-Term Memory (LSTM, **?** and is completely deterministic such that at each time step $t$ the *LSTM* decoder takes as input the previously generated embeddings and the latent code $z$ which captured node-invariant features of the graph. Each embedding is computed

as a function of the concatenation of the current hidden state and the latent code $z$ such that:

$$h_{t+1} = f^d_{LSTM}(g_{in}([z, s_t]), h_t) \tag{4}$$

$$s_{t+1} = f_{embed}([h_{t+1}, z]), \tag{5}$$

where $f^d_{LSTM}$ corresponds to the LSTM recurrence operation and $g_{in}$ and $f_{embed}$ are parametrized as simple MLP layers to perform nonlinear feature extraction.

**Step 4: Graph Decoding from Node Embeddings.** As stated previously, we want to drive the generation of the continuous embeddings $s$ towards latent factors that contains enough information about the node they represent (i.e. we can easily retrieve the one-hot atom type performing a linear transformation of the continuous embedding) and its neighbourhood (i.e. the adjacency tensor can be easily retrieved by comparing those embeddings in a pair-wise manner). For those reasons, we suggest to factorize each bond type that we can have between two atoms in a relational inference fashion (Zitnik et al., 2018; Kipf et al., 2018).

Let $S \in \mathbb{R}^{n \times p}$ be the concatenated continuous node embeddings generated in the previous step. We reconstruct the adjacency tensor $E$ by learning edge-specific similarity measure as follows for $k$-th edge type:

$$p(E_{:,:,k}|S) = \prod_{i=1}^{n} \prod_{j=1}^{n} p(E_{i,j,k}|s_i, s_j). \tag{6}$$

We model this by a set of edge-specific factors $U = (u_1, \cdots, u_e) \in \mathbb{R}^{e \times p}$ such that we can reconstruct the adjacency tensor as :

$$\tilde{E}_{i,j,k} = \sigma(s_i^T D_k s_j) = p(E_{i,j,k}|s_i, s_j), \tag{7}$$

where $\sigma$ is the logistic sigmoid function, $D_k$ the corresponding diagonal matrix of the vector $u_k$ and the factors $(u_i) \in \mathbb{R}^{e \times p}$ are learned parameters.

We reconstruct the node features (i.e. the atom type) with a simple affine transformation such that:

$$\tilde{N}_{i,:} = p(N_i|s_i) = \text{softmax}(W s_i), \tag{8}$$

where $W \in \mathbb{R}^{p \times d}$ is a learned parameter.

These four steps define our proposed graph autoencoder.

**Generating Graphs of Arbitrary Sizes.** In order to generate graphs of different sizes we need to add what we call here an **Existence** module that retrieves a probability of belonging to the final graph from each continuous embedding generated by the embeddings generator (in step 3). This module is parametrized as a simple MLP layer followed by a sigmoid activation and stops the unrolling of the embedding LSTM generator whenever we encounter a *non-informative* embedding. This module can be interpreted as an $< eos >$ translator.

## 3.2 Training

**Teacher forcing.** In order to make the model converge in reasonable time we used a trick similar to the teacher-forcing based training of language models (Williams & Zipser, 1989). The training is thus done in 3 phases:

- We first pre-train the GCN part along with the embedding decoder (factorization, nodes and existence modules) to reconstruct the graphs. This corresponds to the training of a simple Graph AE as in Kipf & Welling (2016a) except that we also want to reconstruct the nodes' one-hot features (and not just the *relations*).

- We then append those two units to the embedding aggregator and generator while keeping them fixed. In this second phase, the embedding generator is trained in a teacher forcing fashion where at each time step $t$ the *LSTM* decoder does not take as input the previously generated embedding but the *true* one that is the direct output of the pretrained GCN embedding encoder.

- Finally in order to transition from a teacher-forcing to a fully autoregressive state we increasingly (Bengio et al., 2015) feed the LSTM generator more of its own predictions. When that fully autoregressive state is reached the pre-trained units are not frozen anymore and the whole model continues training end-to-end.

**Log-Likelihood Estimates**  We train this first autoencoder framework in a MLE fashion with the following negative log-likelihood estimate $\mathcal{L}_{rec} = \mathcal{L}_X + \mathcal{L}_{\bar{X}} + \mathcal{L}_N$ corresponding to the existing edges ($X$), the non-existing edges ($\bar{X}$) and the node features ($N$) reconstruction terms:

$$\mathcal{L}_X = -\frac{1}{|X|} \sum_{(i,j) \in X} E_{i,j,:}^T \log(\tilde{E}_{i,j,:}) + (1 - E_{i,j,:})^T \log(1 - \tilde{E}_{i,j,:}) \tag{9}$$

$$\mathcal{L}_{\bar{X}} = -\frac{1}{|\bar{X}|} \sum_{(i,j) \in \bar{X}} \sum_k \log(1 - \tilde{E}_{i,j,k}) \tag{10}$$

$$\mathcal{L}_N = -\frac{1}{n} \sum N^T \log(\tilde{N}), \tag{11}$$

where $X$ is the set of existing edges, $\bar{X}$ the set of non existing edges, $E$ the adjacency tensor, $N$ the node features tensor, $n$ the number of nodes in the graph. As molecular graphs are sparse we found that such separate normalisations were crucial for the training.

## 3.3 Conditional Generation and Optimisation

**Model overview.**  In this part we discuss our approach to build a conditional framework starting from the previous autoencoder architecture where the construction of a probabilistic graph $\tilde{G}$ is conditioned on some unregularized latent code $z$ derived from a given input graph. We then augment this unstructured $z$ with a set of structured attributes $y$ that represent some physico-chemical properties of interest so that the decoder is conditioned on the joint $(z, y)$. At the end of a successful training we expect this decoder to generate samples that have the properties specified in $y$ and to be *similar* (in terms of information contained in $z$) to the original query molecular graph (encoded as $z$). To do so we choose a mutual information maximization approach (detailed in the appendix) that involves the use of discriminators that assess the properties $\tilde{y}$ of the generated samples.

**Discriminator Pre-Training**  In this phase we pre-train a discriminator to assess the property $y$ of a generated sample so that we can backpropagate its feedback to the generator (the discriminator can be trained on another dataset and we can have several discriminators for several attributes of interest). In order to have informative gradients in the early stages of the training we have trained the discriminator on continuous approximations of the discrete training graphs (details in the appendix A.1) so that our objective becomes:

$$\mathcal{L}_{dis} = \mathbb{E}_{(x,y) \sim \tilde{p}_{data}(x,y)}[-\log Q(y|x)], \tag{12}$$

where the graphs sampled from $\tilde{p}_{data}(x)$ are the probabilistic approximations of the discrete ones.

The next step is to incorporate the feedback signal of the trained discriminator in order to formulate the property attribute constraint. The training is decomposed in two phases in which we learn to reconstruct graphs of the dataset (**MLE** phase) and to modify chemical attributes (**Variational MI maximization** phase).

| Model | Reconstruction |
|---|---|
| JT-VAE (with stereochemistry) | 76.7 |
| JT-AE | 69.9 |
| DEFactor - 56 | 89.2 |
| DEFactor - 64 | 89.4 |
| **DEFactor - 100** | **89.8** |

Table 1: **Molecular graph reconstruction task**. JT-VAE result is taken from Jin et al. (2018) and uses a latent code of size 56. We compared the expressivity of our decoder in a molecular reconstruction task with the JT-VAE decoder. JT-VAE model refers to the original VAE framework as described in in Jin et al. (2018) whereas JT-AE is the adapted deterministic version of it (we removed the stereochemistry information of the molecules as we believe it is an unnecessary burden for the model).

**Encoder Learning.** The encoder is updated only during the reconstruction phase where we sample attributes $y$ from the true posterior. The encoder loss is a linear combination of the molecular graph reconstruction ($\mathcal{L}_{rec}$) and the property reconstruction( $\mathcal{L}_{prop}$ ) s.t. $\mathcal{L}_{rec} = \mathbb{E}_{(x,y)\sim p_{data}(x,y),z\sim E(z|x)}[-\log p_{gen}(x|z,y)]$ (using the log-likelihood estimates in **(7)**) and $\mathcal{L}_{prop} = \mathbb{E}_{(x,y)\sim p_{data}(x,y),z\sim E(z|x),x'\sim p_{gen}(x|z,y)}[-\log Q(y|x')]$. The total encoder loss is:

$$\mathcal{L}_{enc} = \mathcal{L}_{rec} + \beta\mathcal{L}_{prop}. \tag{13}$$

**Generator Learning.** The generator is updated in both reconstruction and conditional phases. In the **MLE** phase the generator is trained with same loss $\mathcal{L}_{enc}$ as the encoder so that it is pushed towards generating realistic molecular graphs. In the **MI maximization** phase we sample the attributes from a prior $p(y)$ s.t. we minimize the following objective: $\mathcal{L}_{cond} = \mathbb{E}_{x\sim p_{data}(x),y\sim p(y)z\sim E(z|x),x'\sim p_{gen}(x|z,y)}[-\log Q(y|x')]$,

$$\mathcal{L}_{gen} = \mathcal{L}_{rec} + \alpha\mathcal{L}_{cond} + \beta\mathcal{L}_{prop}. \tag{14}$$

In this phase the only optimisation signal comes from the trained discriminator. Consequently there is a risk of falling off the manifold of the molecular graphs as no *realism* constraint is specified. A good way to make sure that this does not happen is to add a similar discriminator trained to distinguish between the real probabilistic graph and the generated ones so that when trying to satisfy the attribute constraint the generator is forced to produce valid molecular graphs. We leave that additional feature for future work.

## 4 EXPERIMENTS

To compare with recent results in constrained molecular graph optimization Jin et al. (2018); You et al. (2018a), we present the following experiments :

- **Molecular Graph Reconstruction**: We test the autoencoder framework on the task of reconstructing input molecules from their latent representations.

- **Conditional Generation**: We test our conditional generation framework on the task of generating novel molecules that satisfy a given input property. Here, we are interested in the octanol-water partition coefficient (LogP) optimization used as benchmark in Kusner et al. (2017); Jin et al. (2018); You et al. (2018a).

- **Constrained Property Optimization**: Finally, we test our conditional autoencoder on the task of modifying a given molecule to improve a specified property, while constraining the degree of deviation from the original molecule. Again we use the LogP benchmark for the experiment.

**Molecular graph reconstruction:**   In this task we evaluate the exact reconstruction error from encoding and decoding a given molecular graph from the test set. We report in Table 1 the ratio of exactly reconstructed graphs, where we see that the our autoencoder outperforms the Junction Tree-VAE (JT-VAE) Jin et al. (2018) which has the current state-of-the-art performance in this task. Appendix B.3 reports the reconstruction ratio as a function of the molecule size (number of heavy atoms).

**Conditional Generation:**   In this task we evaluate the conditional generation formulation described in Section 3.3. For a given molecule $m_i$ with an observed property value $y_i$, the goal here is to modify the molecule to generate a new molecule with the given target property value; $(m_i', y_i^*)$.

New molecules are generated by conditioning the decoder on $(z_i; y_i^*)$, where $z_i$ is the latent code for $m_i$. The decoded new molecule $m_i'$, is ideally best suited to satisfy the target property. This is evaluated by comparing the property value of the new molecule with the target property value. A generator that performs well at this task will produce predicted molecules with property values that are close to the target. In these experiments, LogP was chosen as the desired property, and we use  RDKit, online to calculate the LogP values of generated molecules.

The scatter plots in Figure 2 give for a randomly selected set of test molecules, the correlation of target property values against the evaluated property value of the correctly decoded molecules.

**Constrained Property Optimization:**   In this section we follow the evaluation methodology outlined in Jin et al. (2018); You et al. (2018a), and evaluate our model in a constrained molecule property optimization. In contrast to Jin et al. (2018), because of the conditional formulation, our autoencoder is already suited for that task without the need for retraining.

Given the 800 molecules with the lowest penalized LogP[1] property scores from the test set, we evaluate the decoder by providing pairs of $(z_i, y_i^*)$ with increasing property scores, and among the valid decoded graphs we compute:

- Their similarity scores (Sim.) to the encoded target molecule (called the prototype);

- Their penalized LogP scores. Note that in this setting the conditioning property values $(y_i^*)$ are the unpenalized LogP scores. However, to evaluate the model we compute the penalized LogP scores to assess the model's ability to decode synthetically accessible molecules.

- While varying the similarity threshold values $(\delta)$, we compute the success rate (Suc.) for all 800 molecules. This measures how often we to get a novel molecule with an improved penalized LogP score.

- Finally, for different similarity thresholds, for successfully decoded molecules, we report the average improvements (Imp.) and the similarity (Sim.) for the molecule that is most improved. We compare our results with Jin et al. (2018); You et al. (2018a).

The final results are reported in Table 2. As can be seen, although slightly behind GCPN You et al. (2018a) w.r.t. success rates (Suc.), DEFactor significantly outperforms other models in terms of improvements (Imp.) achieved (by between 130% and 195% for thresholds 0.2 and 0.6 respectively, with respect to the next best model GCPN).

## 5    Future work

In this paper, we designed a new way of generating molecular graphs in a conditional optimisation setting . We believe that our *DEFactor* model is a significant step forward

---

[1]The penalized logP is octanol-water partition coefficient (logP) penalized by the synthetic accessibility (SA) score and the number of long cycles, see Jin et al. (2018)

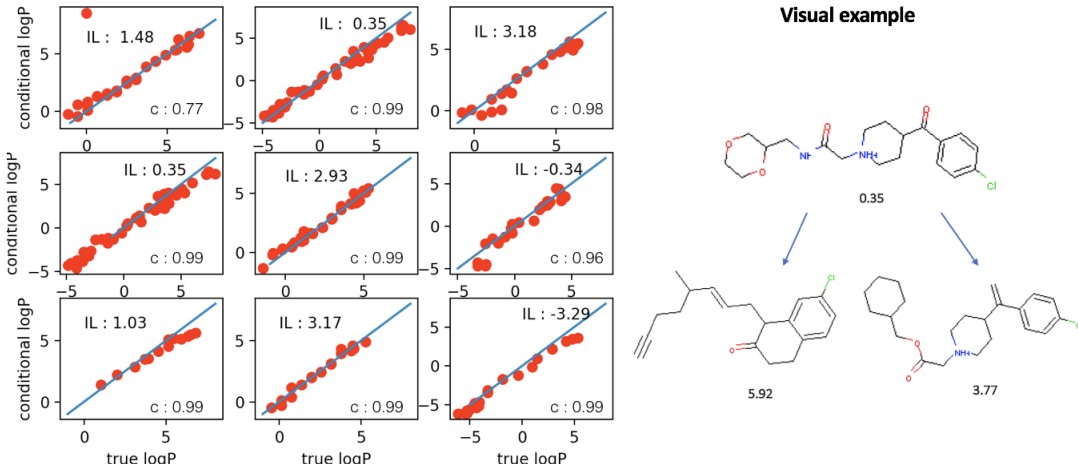

Figure 2: **Conditional generation**: The initial LogP value of the query molecule is specified as *IL* and the Pearson correlation coefficient is specified as *c*. We report on the y-axis the conditional value given as input and on the x-axis the true LogP of the generated graph when translated back into molecule (we have of course only reported the values when the decoded graphs correspond to valid molecules, which explains the difference in the total number of points for each query molecule).

| $\delta$ | JT-VAE | | | GCPN | | | DEFactor | | |
|---|---|---|---|---|---|---|---|---|---|
| | Imp. | Sim. | Suc. | Imp. | Sim. | Suc. | Imp. | Sim. | Suc. |
| **0.0** | 1.91± 2.04 | 0.28±0.15 | 97.5% | 4.20±1.28 | **0.32±0.12** | **100**% | **6.62±2.50** | 0.20±0.16 | 91.5% |
| **0.2** | 1.68± 1.85 | 0.33±0.13 | 97.1% | 4.12±1.19 | **0.34±0.11** | **100**% | **5.55±2.31** | 0.31±0.12 | 90.8% |
| **0.4** | 0.84± 1.45 | **0.51±0.10** | 83.6% | 2.49±1.30 | 0.47±0.08 | **100**% | **3.41±1.8** | 0.49±0.09 | 85.9% |
| **0.6** | 0.21± 0.71 | 0.69±0.06 | 46.4% | 0.79±0.63 | 0.68±0.08 | **100**% | **1.55±1.19** | **0.69±0.06** | 72.6% |

Table 2: **Constrained penalized LogP maximisation task**: each row shows a different level of similarity constraint $\delta$ and columns are for improvements (Imp.), similarity to the original query (Sim.), and the success rate (Suc.). Values for other models are taken from You et al. (2018a).

to build ML-driven applications for de-novo drug design or generation of molecules with optimal properties without resorting to methods that do not directly optimise the desired properties.

Note that a drawback of our model is that it uses an MLE training process which forces us to either fix the ordering of nodes or to perform a computationally expensive graph matching operation to compute the loss. Moreover in our fully deterministic conditional formulation we assume that chemical properties optimisation is a one-to-one mapping but in reality there may exist many suitable way of optimizing a molecule to satisfy one property condition while staying similar to the query molecule. To that extent it could be interesting to augment our model to include the possibility of a one-to-many mapping. Another way of improving the model could also be to include a validity constraint formulated as training a discriminator that discriminates between valid and generated graphs.

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

## Appendix A    Models comparison

| Model | Inference | Parameters | Constrained | Probabilistic | No Retraining |
|-------|-----------|------------|-------------|---------------|---------------|
| MolGAN | ✗ | ✗ | ✗ | ✓ | NA |
| JT-VAE | ✓ | ✓ | ✓ | ✗ | ✗ |
| GCPNN | ✗ | ✓ | ✓ | ✗ | ✓ |
| DEFactor(Ours) | ✓ | ✓ | ✓ | ✓ | ✓ |

Figure 3: We report here a comparison of the abilities of previous recent models involving molecular graph generation and optimization

We are interested in the following features of the models :

- **Inference** : If the model is equipped or not with an inference network. To encode some target molecule like we do in the conditional setting.

- **Parameter-efficient** : If the number of parameters of the model depends on the graph sizes.

- **Constrained** : If the model is studied in a constrained optimization scenario : namely the case where we want to optimize a property while constraining the degree of deviation from the original molecule.

- **Probabilistic** : If the outptut of the model is a probabilistic graph s.t. it is differentiable w.r.t to the decoder's parameters.

- **No Retraining** : If we need to retrain/fine-tune/perform gradient-ascent each time we want to optimize a novel molecule.

## Appendix B    Conditionnal setting

### B.1    Graphs continuous approximation

For the pre-training of the discriminators we suggested to train them on continuous approximation of the discrete graphs that *ressembles* the output of the decoder. To that extent we used the trained partial graph autoencoder (used for the teacher forcing at the beginning of the training of the full autoencoder)

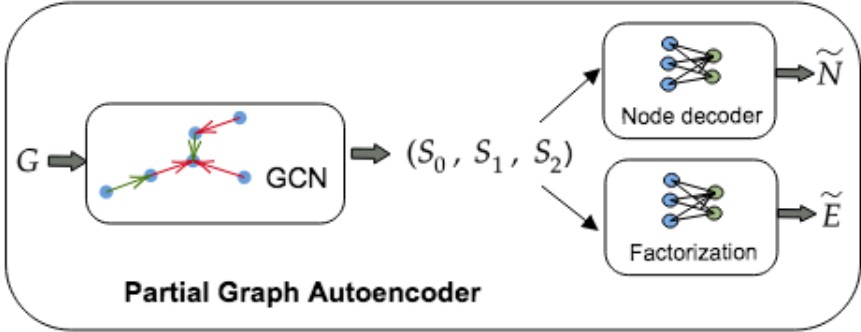

Figure 4: Partial graph Autoencoder used for the pre-training part

### B.2 MUTUAL INFORMATION MAXIMIZATION

For the conditional setting we choose a simple mutual information maximization formulation. The objective is to maximize the MI $I(X;Y)$ between the target property $Y$ and the decoder's output $X = G_\theta(Y)$ under the joint $p_\theta(X,Y)$ defined by the decoder $G_\theta$. In the conditional setting $G_\theta$ is also conditioned on the encoded molecule $z$ but for simplicity we treat it as a parameter of the decoder (and thus reason with one target molecule from which we want to modify attributes). We define the MI as:

$$
\begin{aligned}
I(y;G_\theta(y)) &= \mathbb{E}_{x\sim G_\theta(y)}[\mathbb{E}_{y\prime\sim p_\theta(y|x)}[\log p_\theta(y\prime|x)]] + H(y) \\
&= \mathbb{E}_{x\sim G_\theta(y)}[D_{KL}(p_\theta(.|x)||Q(.|x)) \\
&\quad + \mathbb{E}_{y\prime\sim p_\theta(y|x)}[\log Q(y\prime|x)]] + H(y) \\
&\geq \mathbb{E}_{x\sim G_\theta(y)}[\mathbb{E}_{y\prime\sim p_\theta(y|x)}[\log Q(y\prime|x)]] + H(y)
\end{aligned}
$$

In our conditional setting we pre-trained the discriminators (parametrized by $Q$ in the lower bound derivation) to approximate $p_{data}(y|x)$ which makes the bound tight only when $p_\theta(y_{paired}|x)$ is close to $p_{data}(y|x)$ and this corresponds to a stage where the decoder has maximized the log-likelihood of the data well enough (i.e. when it is able to reconstruct input graphs properly when $z$ and $y$ are paired). Thus, in the conditional setting we are maximizing the following objective:

$$
\mathcal{L}_{cond} = \mathbb{E}_{x,y\sim p_{data}(x,y),z\sim E(x),y\prime\sim p(y)}[\log G_\theta(y,z) + I(y\prime;G_\theta(y\prime,z))]
$$

### B.3 RECONSTRUCTION AS A FUNCTION OF NUMBER OF ATOMS

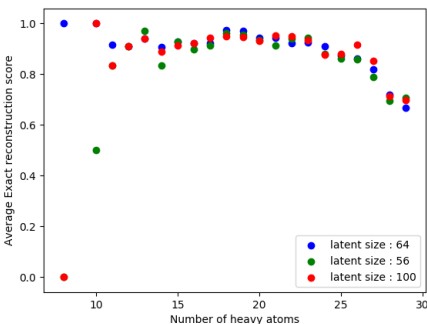

Figure 5: Accuracy score as a function of the number of heavy atoms in the molecule(x axis) for different size of the latent code

Notice that as we make use of a simple LSTM to encode a graph representation, there is a risk that for the largest molecules the long term dependencies of the embeddings are not captured well resulting in a bad reconstruction error. We capture this observation in figure 4. One possible amelioration could be to add other at each step other non-sequential aggregation of the embeddings (average pooling of the emebeddings for example) or to make the encoder more powerful by adding some attention mechanisms. We leave those for future work.

### B.4 VISUAL SIMILARITY SAMPLES

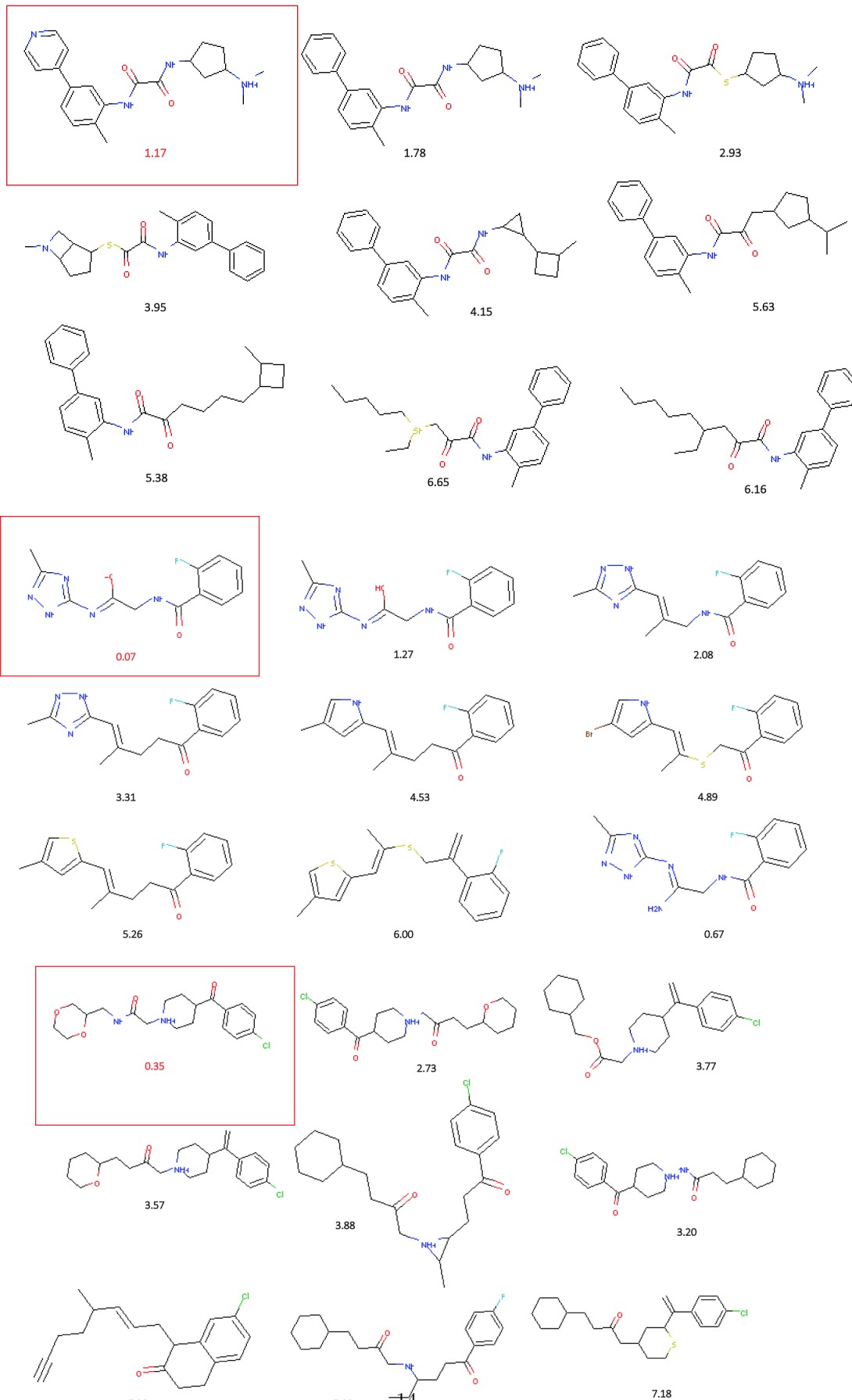

Figure 6: LogP increasing task visual example. The original molecule is circled in red.