# OpenReview forum: "DEFactor: Differentiable Edge Factorization-based Probabilistic Graph Generation"
_ICLR.cc/2019/Conference_

### Official Review · AnonReviewer3 · 2018-11-02
**Experiments and Writing Need Improvement**

**Rating:** 4
**Confidence:** 4

**Review:**

In this paper, authors propose a deep generative model and a variant for graph generation and conditional graph generation respectively. It exploits an encoder which is built based on GCN and GraphSAGE, a autoregressive LSTM decoder which generates the graph embedding, and a factorized edge based probabilistic model for generating edge and node type. For conditional generation, authors also propose a discriminating training scheme based on maximizing the mutual information. Experiments on ZINC dataset show that the proposed method is promising.

Strength:

1, The problem this paper tries to tackle is very challenging and of great significance. Especially, the conditional graph generation direction under the deep learning context is novel.

2, The overall model is interesting although it is a bit complicated as it combines quite a few modules.

Weakness:

1, In the reconstruction experiment, comparisons with several recent competitive methods are missing. For example, the methods which have been already discussed in the related work, Li et al. (2018a), You et al. (2018a) and You et al. (2018b). Moreover, it is not explained whether the comparison setting is the same as Jin et al. (2018) and what the size of the latent code of their method is. It seems less convincing by just taking results from their paper and do the comparison.

2, Authors motive their work by saying in the abstract that “other graph generative models are either computationally expensive, limiting their use to only small graphs or are formulated as a sequence of discrete actions needed to construct a graph, making the output graph non-differentiable w.r.t the model parameters”. However, if I understood correctly, in Eq. (7), authors compute the soft adjacency tensor which is a dense tensor and of size #node by #node by #edge types. Therefore, I did not see why this method can scale to large graphs.

3, The overall model exploits a lot of design choices without doing any ablation study to justify. For example, how does the pre-trained discriminator affect the performance of the conditional graph generation? Why not fine-tune it along with the generator? The overall model has quite a few loss functions and associated weights of which the values are not explained at all.

4, Conditional generation part is not written clearly. Especially, the description of variational mutual information phase is so brief that I do not understand the motivation of designing such an objective function. What is the architecture of the discriminator?

5, How do authors get real attributes from the conditionally generated molecules? It is not explained in the paper.

Typos:

1, There are a few references missing (question mark) in the first and second paragraphs of section 2.

2, Methods in the experiment section are given without explicit reference, like GCPN.

3, Since edge type is introduced, I suggest authors explicitly mention the generated graphs are multi-graph in the beginning of model section.

Overall, I do not think this paper is ready for publishing and it could be improved significantly.

---------------------------------------------------------------------------------------------------------------------------------------------------------------------

Update:

Thanks for the detailed explanation. The new figure 1 is indeed helpful for demonstrating the overall idea.

However, I still found some claims made by authors problematic.
For example, it reads in the abstract that "...or are formulated as a sequence of discrete actions needed to construct a graph, making the output graph non-differentiable w.r.t the model parameters...".
Clearly, Li et al. 2018b has a differentiable formulation which falls under your description.

Besides, I suggest authors adjust the experiment such that it focuses more on comparing conditional generation.
Also, please set up some reasonable baselines based on previous work rather than saying it is not directly comparable.
Directly taking numbers from other papers for a comparison is not a good idea given the fact that these experiments usually involve quite a few details which could potentially vary significantly.

Therefore, I would like to keep my original rating.

---

> ### Author Response · Authors · 2018-11-27
> **Response to reviewer 3 [part 1]**
>
> We thank the reviewer for his useful comments and will address his concerns sequentially.
>
> 1. For the comparison with Jin et Al [1], we recomputed the results in the same settings as ours in the revised version : deterministic AE, 2D molecular graphs, 56 latent size.
>
> Concerning the missing comparison with the competitive methods suggested by the reviewer we would like to clarify that :
>          - We do discuss those methods in the related work part as existing generative models on molecular graphs but they were not designed to reconstruct a particular graph, and though are not adapted to an exact reconstruction task.
>          - In fact those methods do not come with an inference network whose output can be used to condition the whole generation process to match the condition.
>
> Let us suppose that we equip those generative models with an encoder (typically ours), their decoding schemes involve a prediction of very long sequences of action and coming with a good training procedure in that context is not trivial at all  :
>           - How do we choose the sequences order ? GCPN (and others) actually argues that using a fixed order as the domain suggested one (SMILES) yields overfitting of their model. To that end they make use of randomly selected state transition in the training set. Naturally this training procedure is not applicable to the task of exact graph reconstruction (where we want to reconstruct the exact full sequence of actions).
>          - Li et al actually states : “ The generation process used by the graph model is typically a long sequence of decisions. If other forms of graph linearization is available, e.g. SMILES, then such sequences are typically 2-3x shorter. This is a significant disadvantage for the graph model, it not only makes it harder to get the likelihood right, but also makes training more difficult.[...]We have found that training such graph models is more difficult than training typical LSTM models. The sequences these models are trained on are typically long, and the model structure is constantly changing, which leads to unstable training. Lowering the learning rate can solve a lot of instability problems, but more satisfying solutions may be obtained by tweaking the model.”
>
> We overall think that redesigning the suggested models so that they can perform well on an exact reconstruction task is not trivial and would constitute potentially completely different models in the end. Spending considerable amount of time  trying to repurpose existing generative models does not seem reasonable. The very example of the JT-VAE that was designed (by its architecture) to reconstruct graphs exactly support our opinion : we tried to go from the probabilistic VAE framework to the deterministic AE one and the best reconstruction result we could get with an available code was lower that the reported one on the VAE setting.
>
> 2. We  agree on the misleading for use of ‘scalable’ and ‘cheap’ in the manuscript.. Actually it was supposed to be understood as it is defined in the original related work section ( “ Scalable :  this means that the number of parameters of the decoder should not depend on a fixed predefined maximum graph size”  like it is the cas for [2] and [3]). We fixed the misuse in the manuscript.
>
> Concerning the large graph statement, the model’s focus is on molecular graphs (which we find is an important problem on its own) thus “large” and “small” do not have the same signification here when compared to general graphs/ networks. Small = less than 10 heavy atoms (like in [2] and [3], they specify small in their title) Large = around 60 heavy atoms which is large enough in the optimization tasks we are interested in the drug discovery pipeline.

---

> > ### Author Response · Authors · 2018-11-27
> > **Response to reviewer 3 [part 2]**
> >
> > 3 and 4. The discriminator is not the central part of the model. In fact it is used for the formulation of the conditional setting, which could be justified/criticized  as way to perform explicit conditional generation on another manuscript proposal. The main purpose of our manuscript is the design of a new graph decoding scheme that allows us to do direct graph optimization without suffering from high variance of REINFORCE-like update. As a proof of the representational power of the decoder we test it in the same constrained property optimization framework than previous competitive models ( namely, JT-VAE and GCPN).
> >
> > Although conditional generation in itself is a very interessant topic we had to make a choice and as we suggest a new model to perform the task we put the emphasis on the description of the model itself rather than the conditional generation formulation.
> >
> > However there is a reason why we do not want to fine-tune the discriminator along with the generator.  Doing so would be actually equivalent to the InfoGAN framework where we allow the model to discover the semantic of the structured part of the latent factors. In our case we do not need to discover the semantics present in our data as we already know what they mean (we know that we want to minimize logP of the molecule and not some discovered semantic whose prior would match the one we are sampling from). To be more precise, if we consider fine-tuning the discriminator along with the generator, the bottleneck would be the early stages of the training (when the generator is pretty bad) and where there is a high risk of observing a “semantic-drift”. Additionally, as the discriminator will be far away from p_data(y|x) the two signals (MLE and VMI maximization) will drive the generator to update its parameters to completely different directions which would make the training harder.  On the contrary when the discriminator has already approximated p_data(y|x) (ie. the final and wanted posterior p_gen(y|x)), the two signals are pointing the generator to the same direction and makes the training easier (and more intuitive).
> >
> > We will add a further detailed section in the appendices to motivate such formulation.
> >
> > 5. In order to test the conditional setting we chose an attribute that we can compute for all the molecules. The way it is computed (in a black-box way through rdkit Library) depends on the fragments present in molecule and is based on some experimental data extensively collected and stored.  Computed via rdkiit’s Crippen function [Landrum, 2016]. We added that in the revised version.
> >
> > We thank the reviewer again and hope we have addressed his concerns in the revised version.
> >
> > The authors
> >
> > [1] Jin et al, Junction Tree Variational Autoencoder for Molecular Graphs Generation, https://arxiv.org/pdf/1802.04364.pdf
> > [2] De Cao & Kipf,  MolGAN : An implicit generative model for small molecular graphs, https://arxiv.org/abs/1805.11973
> > [3] Simonovsky & Komodakis, GraphVAE : Towards Generation of Small Graphs Using Variational Autoencoders, https://arxiv.org/abs/1802.03480

---

### Official Review · AnonReviewer2 · 2018-11-02
**review on "DEFactor: Differentiable Edge Factorization-based Probabilistic Graph Generation"**

**Rating:** 5
**Confidence:** 3

**Review:**

This paper proposed a variant of the graph variational autoencoder [1] to do generative modeling of graphs. The author introduced an additional conditional variable (e.g., property value) into the decoder. By backpropagating through the discriminator, the model is able to find the graph with desired property value.

Overall the paper reads well and is easy to follow. The conditional generation of graphs seems also helpful regarding the empirical performance. However, there are several concerns regarding the paper:

1) The edge factorization-based modeling is not new. In fact [1] already uses the node embeddings to factorize the adjacency matrix. This paper models extra information including node tags and edge types, but these are not fundamental differences compared to [1].

2) The paper claims the method is ‘cheaper’ and ‘scalable’. Since essentially the computation cost is similar to [1] which requires at least O(n^2) to generate a graph with n nodes, I’m not super confident about the author’s claim. Though this can be parallelized, but the memory cost is still in this order of magnitude, which might be too much for a sparse graph. Also there’s no large graph generative modeling experiments available.

3) Continue with 2), the adjacency matrix of a large graph (e.g., graph with more than 1k nodes) doesn’t have to be low rank. So modeling with factorization (with typically ~256 embedding size) may not be suitable in this case.

Some minor comments:
4) Regarding Eq (2), why the lstm is used, instead of some simple order invariant aggregation?

5) the paper needs more refinement. E.g., in the middle of page 2 there is a missing citation.

[1]  Kipf & Welling, Variational Graph Auto-Encoders, https://arxiv.org/pdf/1611.07308.pdf

---

> ### Author Response · Authors · 2018-11-07
> **We think the novelty of the model has been misunderstood. We clarify that point**
>
> We thank the reviewer for the detailed and useful comments. We will proceed to the rebuttal as follows :
>
>     - Specific answers/clarifications on issues raised by the reviewer sequentially
>     - Summary of clarifications made in the answer
>     - Summary of changes in the manuscript implied by the review
>
>  ----- SEQUENTIAL: Clarifications/Answers -----
>
> Our Answer  on 1) Novelty of the model  :
> As referenced in  step 4 in section 3.1, we utilise the edge-factorization described in [1] (VGAE).  The goal here is to generate graphs of varying size given some input condition.  In practice this means being to generate both the nodes and edges of the graph, conditional on some latent  code.
>
> In contrast the VGAE has been designed in the context of relational inference (eg. link prediction in citation network), where the number of nodes is fixed, and task is to learn a suitable representation for the nodes, such that we’re able to reconstruct/predict missing links.
>
>  Because of the assumptions of this setting the VGAE only solves half of the problem we are trying to address: given a set of node embeddings [1] reconstructs the adjacency tensor. In contrast we want to be able to generate both the node embeddings whose number is unknown a priori and their adjacency tensor given some latent code.
>
> In practice this is achieved by adding a new component (see step 3. Sec 3.1) to model how to go from a latent code z to an actual set of node embeddings.That specific node embeddings generator (that we parametrize with an LSTM)  is  the major contribution of our model.
>
> To clarify the differences between [1] and our DEFactor  we added a short paragraph in the related work section on edge-factorization.
>
>
> Our Answer  on 2) and 3)  Clarification on "large" "cheap" and  "scalable":
> On the use of “scalable”  and “cheap” In the manuscript we state that scalable ``[..] means that the number of parameters of the decoder should not depend on a fixed pre-defined maximum graph size.`` and the number of parameters in DEFactor is independent of the (max) size of the graph unlike  [2] and [3]. We fixed the misleading use in the manuscript
>
> On the large graph modeling concern : the model’s focus is on molecular graphs (which we think is an important problem on its own) thus “large” and “small” do not have the same signification here when compared to general graphs/ networks (that is why we put large in italic style in the first version of the manuscript in the bullet points of the related work section  but we updated it into “large molecular graphs”) . Small = less than 10 heavy atoms (like in [2] and [3], they specify small in their title) Large = around 60 heavy atoms which is large enough in the optimization tasks we are interested in the drug discovery pipeline.
>
>
>
>                    4) Minor comments , Reviewer :  “Regarding Eq (2), why the lstm is used, instead of some simple order invariant aggregation?the paper needs more refinement. E.g., in the middle of page 2 there is a missing citation.”
>
> We actually tried the simpler order invariant aggregation function (avg and max)  but the convergence was bad so we directly went for a more complex/richer feature extractor such as LSTM.  We agree that concerns can be raised concerning the matter of the order when we use such sequential aggregation functions however (as specified in section 3.1 (step 1 and 2) ) we trained the LSTM with a randomly permuted order of the embeddings it has to encode and did not notice any change in the performance of the model.
>
> Broken link fixed, thanks :)
>
>
> -----  SUMMARY : What we think  we clarified ------
>
> - The  novelty of the proposed decoder ( = autoregressive generation of nodes embeddings for graphs of varying size) which we think has been misunderstood by the reviewer.
> - The misleading use of words “scalable” and “cheap” : we meant only meant that the number of parameters should not depend on the graph size (when compared to [2] and [3]).
> - The meaning of “large” graphs in the context of molecular graphs (which we find is a relevant and important problem on its own).
>
> ---- ACTION POINTS : What we modified in the manuscript ----
>
> - Corrected the misleading use of “scalable” and “cheap” in the manuscript
> - Replace large graphs by large molecular graphs to specify the scale of graphs we are referring to
> - Added a paragraph in the related work section on the edge-factorization to further emphasize the true novelty of our decoder.
> - Fixed the broken references
>
> We hope that our answers clarified our contribution and thank the reviewer again,
>
> The Authors
>
> --- REFERENCES ---
>
> [1] Kipf & Welling, Variational Graph Auto-Encoders , https://arxiv.org/pdf/1611.07308.pdf
> [2] De Cao & Kipf,  MolGAN : An implicit generative model for small molecular graphs, https://arxiv.org/abs/1805.11973
> [3] Simonovsky & Komodakis, GraphVAE : Towards Generation of Small Graphs Using Variational Autoencoders, https://arxiv.org/abs/1802.03480

---

### Official Review · AnonReviewer1 · 2018-11-09
**The paper is very poor--- not ready for publication**

**Rating:** 3
**Confidence:** 5

**Review:**

The paper proposes a conditional graph generation that directly optimizes the properties of the graph. The paper is very weak.
1. I think almost all probabilistic graph generative models are differentiable. If the  objective is differentiable function of real
    variables, it is usually differentiable.

2.  The authors claim that existing works Simonovsky and Komodakis (2018) and Cao & Kipf (2018) are restricted to use small graphs with predefined maximum size. This work does not overcome the limitation of small graphs issue too.

3. The authors do not show any measure on validity, novelty or uniqueness which are now standard in literature.
   Also I do not find any comparison with molGAN paper which tackles a similar objective.

4. Could the authors show if the decoding process is permutation invariant? I am not really sure of that. I was trying to prove that thing formally, but I failed.

---

> ### Author Response · Authors · 2018-11-17
> **Answer to Reviewer 1**
>
> We thank the reviewer for his comments and will answer to these one by one.
>
> 1.
> We also state in the paper that probabilistic graph generative models are differentiable. In the abstract “ In  this  work  we  propose  a  model  for  conditional  graph  generation  that directly  optimises  properties  of  the  graph,  and  generates  a  probabilistic graph,  making  the  decoding  process  differentiable”
>
>
> 2.
> They are limited to very small graph because of their parametriazations : the number of parameters depends on the predefined maximum graph size they have set. If their last hidden layer is of size d, the number of edges r, and the maximum graph size of size n then the weight matrix mapping the last layer to the edge tensor  will be of size n*n*r*d which is very limiting.
>
> Our factorization model does not have this limitation (the number of parameters only depends on the size of the embeddings we choose), and keeping in  memory of the full probabilistic graph is not an issue when working with molecules -> we are talking here of graphs with a maximum number of heavy atoms around 100. So we overcome this limitation by having a model whose number of parameters does not depend on the maximum size of the graph. We will change the manuscript to be more precise regarding that point.
>
>
> 3.
> Those measures are standard for purely generative models (where the task is to generate molecules without other objective, and the molecules are sampled form the prior). Let us cite JT-VAE’s description of the reconstruction task to that extent :  “ 	We test the VAE models on the task of reconstructing input molecules from their latent representations, and decoding valid molecules when sampling from prior distribution. “
>
> Our model is a conditional autoencoder, which is a new setting (we do not put any prior on the latent code).
>
> MolGAN does not tackle the constrained optimization scenario at all and its formulation is not easily transferable to that setting : MolGAN is an implicit generative model. One way to constrain the generation process could be to add a reward signal computing the similarity between the generator output and the query molecule (the prototype) but :
> This would mean retraining/fine-tuning the model for each query molecule
> The constrained scenario would be explicit : which is not the case of other models (JT-VAE, GCPN and ours) in which the similarity constraint is not directly specified. JT-VAE finetunes the encoded representation, GCPN uses the prototype as a starting point, and we used a conditional formulation without retraining needed so such a comparison would be difficult and unfair anyway.
>
> We will add in the appendix a comparative table of previous models to that extent and to justify the comparison effectively made in our manuscript. To the best of our knowledge, only JT-VAE and GCPNN are comparable models in the implicitly constrained optimization scenario.
>
> 4.
> We never claimed that the decoding process  was permutation invariant. We only made sure that we can make the encoder robust to permutations by training it on different permutations of the embeddings it encodes. However the decoder is trained to match the domain canonical order (heavy atoms are ordered as they appear in their SMILES canonical representation).
>
> We thank the reviewer again and  hope our comments shed a clearer light on our manuscript.
>
> The authors

---

### Public Comment · (anonymous) · 2018-10-05
**Table 1 reconstruction accuracy comparison is totally unfair**

Dear authors,

I believe the reconstruction accuracy comparison in Table 1 is totally unfair. First, all the baseline models (CVAE, GVAE, SD-VAE and JT-VAE) are variational autoencoders, and they computed the reconstruction accuracy by encoding the input molecule with stochastic noises. That is,  the latent encoding of x is sampled from the approximate posterior Q(z|x) (which is a Gaussian). It is a stochastic encoding rather than deterministic.
However, the proposed model in this paper is an autoencoder, and the authors computed the reconstruction accuracy using the deterministic encoding of x. This is the main reason why the proposed model has better performance. In fact, all the baseline models followed Kusner et al.'s evaluation method -- sampling multiple z from Q(z|x) and average the reconstruction accuracy over all stochastic samples.

Second, does the proposed model decodes the stereochemistry (e.g. chirality)? If not, then the comparison is again not under the same scenario. I am asking this because I didn't see any stereochemistry presented in Figure 5 in the appendix. All the baseline models reconstruct both 2D and 3D information of an input molecule. It is important because there is no way to correctly reconstruct a molecule if its stereochemistry is not reconstructed correctly.

I think the authors should remove this problematic comparison, or recompute the accuracy of the proposed model so that all models are compared under the same setting. Table 1 is really misleading, especially to reviewers who are outside of this domain.

---

> ### Author Response · Authors · 2018-10-19
> **On reconstruction accuracy table.**
>
> Thank you for your interest in our article, and sorry for our delayed answer.
>
> --- On your  stereochemistry concern ---
> The way we understood the graph-related prior work (ie. JT-VAE) is that it does NOT reconstruct the 3D structure. However they do evaluate on 3d structures (doing a post-ranking scheme). They actually reconstruct a 2D molecular graph, then list all the possible stereoisomers, rank them and take the most ‘likely’ by computing the cosine similarity score between the encoded molecule and the embedding of all the stereoisomers (see Appendix B  of the JT-VAE article). For that very reason ( ie. the all the stereoisomers are listable given a 2d structure) we believe that working on 2d structure is not only easier but also enough.
>
>
> --- On your stochastic vs deterministic reconstruction concern ---
> For the stochastic VAE vs. deterministic AE we totally agree with you and will add a note to specify this unbalance. However one might argue that it is rather that computing an exact reconstruction score may not be suited to evaluate a VAE model whereas it is a good indicator in our deterministic AE. However we did try to train the JT-VAE model in its AE version on 2d structures and we got a lower reconstruction score than the VAE version reported in the article, which we found weird and did not report it.
>
>
> --- Aim of table 1 ---
> All in all, the major aim of our table 1 is to give a sense of the representative power of our proposed decoder and the comparisons are just here as an indication as, again, the evaluation context is not the same. We will add a comment to clarify those discrepancies between our model and the prior ones.

---

> > ### Public Comment · (anonymous) · 2018-10-19
> > **All methods (especially SMILES based methods) do reconstruct the stereochemistry**
> >
> > Hi,
> >
> > Thanks for your reply. Regarding stereochemistry:
> > 1) SMILES based methods (e.g. SD-VAE) do reconstruct 3D structure (with one step). Therefore, Table 1 is a unfair comparison. As long as you compute the reconstruction accuracy based on "2D structure exact match", Table 1 cannot be right.
> > 2) JT-VAE does multiple steps of generation. However, they still computed the reconstruction accuracy based on "exact match on 3D structures".
> > 3) Is stereochemistry that hard to reconstruct for your model? It's just additional edge labels right?
> >
> > I totally agree that reconstruction accuracy is not a good metric for comparing VAEs. But regardlessly prior work did that comparison, and I am not happy with sloppy experiments. We should really be rigid in experiments and comparisons. After all, I am just trying to help you improve your manuscript. That's it.
> >
> > Thanks again for your reply

---

> > > ### Author Response · Authors · 2018-10-19
> > > **Will update the stereochemistry issue**
> > >
> > > Hi,
> > > Thanks for reply.
> > >
> > > 1 and 3)  Again we believe that adding the stereochemistry is an unnecessary burden but as it is simple to add to the model we will re run the reconstruction task taking into account the 3D structure (like you said it is just additional edge labels to add).
> > > 2) Our main concern with JT-VAE is that we tried to train in its deterministic AE and computed the 'exact match on the 2D structures' but found a lower score than on the 3d stochastic version, which is why we did not report it.
> > >
> > > Thanks again for your comments and clarifications, we will update the manuscript regarding this point.

---

### Meta-Review · Area_Chair1 · 2018-12-13
**the reviewers are unanimous**

**Confidence:** 4
**Recommendation:** Reject

**Metareview:**

Since the reviewers unanimously recommended rejecting this paper, I am also recommending against publication. The paper considers an interesting problem and expresses some interesting modeling ideas. However, I concur with the reviewers that a more extensive and convincing set of experiments would be important to add. Especially important would be more experiments with simple extensions of previous approaches and much simpler models designed to solve one of the tasks directly, even if it is in an ad hoc way. If we assume that we only care about results, we should first make sure these particular benchmarks are difficult (this should not be too hard to establish more convincingly if it is true) and that obvious things to try do not work well.